# Control of Mitochondrial Activity by the Ubiquitin Code in Health and Cancer

**DOI:** 10.3390/cells12020234

**Published:** 2023-01-05

**Authors:** Laura Rinaldi, Emanuela Senatore, Rosa Iannucci, Francesco Chiuso, Antonio Feliciello

**Affiliations:** Department of Molecular Medicine and Medical Biotechnology, University of Naples, 80131 Naples, Italy

**Keywords:** UPS, mitochondria, cancer, dynamics, metabolism

## Abstract

Cellular homeostasis is tightly connected to the broad variety of mitochondrial functions. To stay healthy, cells need a constant supply of nutrients, energy production and antioxidants defenses, undergoing programmed death when a serious, irreversible damage occurs. The key element of a functional integration of all these processes is the correct crosstalk between cell signaling and mitochondrial activities. Once this crosstalk is interrupted, the cell is not able to communicate its needs to mitochondria, resulting in oxidative stress and development of pathological conditions. Conversely, dysfunctional mitochondria may affect cell viability, even in the presence of nutrients supply and energy production, indicating the existence of feed-back control mechanisms between mitochondria and other cellular compartments. The ubiquitin proteasome system (UPS) is a multi-step biochemical pathway that, through the conjugation of ubiquitin moieties to specific protein substrates, controls cellular proteostasis and signaling, removing damaged or aged proteins that might otherwise accumulate and affect cell viability. In response to specific needs or changed extracellular microenvironment, the UPS modulates the turnover of mitochondrial proteins, thus influencing the organelle shape, dynamics and function. Alterations of the dynamic and reciprocal regulation between mitochondria and UPS underpin genetic and proliferative disorders. This review focuses on the mitochondrial metabolism and activities supervised by UPS and examines how deregulation of this control mechanism results in proliferative disorders and cancer.

## 1. Introduction

Mitochondria are fundamental organelles for the maintenance of eukaryotic cellular homeostasis. They are primary involved in ATP production, ion storage and metabolic intermediates production, and represent the principal source of reactive oxygen species (ROS). All these roles place mitochondria at the center stage of cellular health, making them essential for cell differentiation, growth and survival. Mitochondrial fusion and fission, mitochondrial unfolded protein response (UPR^mt^) and mitophagy are important regulatory mechanisms for the maintenance of mitochondrial integrity and activity, and to minimize cellular damages caused by internal or external insults [1,2,3,4].

Evidence has shown the existence of a dynamic crosstalk between the ubiquitin-proteasome system (UPS) and the mitochondrial health [5,6]. Ubiquitination of proteins is an essential mechanism to control a variety of biological functions, such as metabolism, growth, differentiation and development [7,8,9]. A wide variety of E3 ubiquitin ligases, enzymes that transfer ubiquitin molecules from E2-conjugating enzymes to protein substrates, have been identified and functionally characterized [10]. The fate of ubiquitinated proteins depends on the type of ubiquitin modification. Proteins can be mono, poly or multi-mono ubiquitinated. In the mono-ubiquitination reaction a single ubiquitin molecule is added to a specific lysine residue on the target substrate, whereas poly-ubiquitination requires the addition of linear or branched ubiquitin chains to a lysine residue [11]. The position of the lysine residue on the ubiquitin moiety in the poly-ubiquitination reactions defines the fate of the target protein. In fact, ubiquitination involving lysine 11 (K11) or lysine 48 (K48) of ubiquitin are generally linked to proteolysis of substrates, instead ubiquitination involving lysine 63 (K63) are related to non-proteolytic functions, such as regulation of protein localization and/or activity [12,13]. Three different E3 ubiquitin ligases have been identified at the outer mitochondrial membrane (OMM): membrane-associated ring finger 5 (MARCH5/MITOL), mitochondrial ubiquitin ligase activator of NFKB1 (MUL1) and ring finger protein 185 (RNF185) [14,15,16]. Other E3 ligases located in the cytoplasm, such as Seven In-Absentia Homolog 2 (SIAH2) and Parkin, can be also targeted to OMM under certain circumstances to regulate mitochondrial responses to specific metabolic needs or environmental stimuli [17,18]. Ubiquitination is a reversible protein modification since deubiquitinating enzymes (DUBs) or ubiquitin specific proteases (USPs), that are often associated to the E3 ligase complex, counteract the ubiquitination process by removing ubiquitin chains from a given substrate [19,20,21]. Among DUBs, the deubiquitinating enzyme USP30 is specifically localized at OMM where it controls the ubiquitination levels of key mitochondrial targets, antagonizing the biological effects of mitochondrial E3 ligases [22]. Moreover, different E2 conjugating enzymes have also been identified as resident of mitochondrial compartments and their differential expression contributes to the homeostatic regulation of mitochondrial proteins stability and activity [23]. Therefore, dysfunction of this mitochondrial surveillance system controlled by ubiquitination/deubiquitination events is often linked to proliferative disorders, including cancer [24,25,26,27]. We will focus on how UPS controls the mitochondrial health state, and how dysregulation or mutations in this control system are involved in the onset and progression of human proliferative disorders.

## 2. The Ubiquitin Proteasome System (UPS) and Mitochondrial Homeostasis

Mitochondria abundance and activity are fundamental elements underlying the dynamic adaptation of cells to stress conditions. The shape and number of mitochondria are tightly controlled by key biological processes, such as fusion and fission (also known as mitochondrial dynamics) and mitophagy that operate in an interconnected and dynamic way to sustain cellular health and metabolic needs.

### 2.1. Mitochondria Dynamics

Mitochondrial fusion and fission are reversible rearrangements of mitochondrial structure. Mitochondria can fuse or divide their inner and outer membranes, assuming an elongated shape or punctiform pattern and promoting the cellular adaptation to changes in nutrients and oxygen availability, to cope cellular stress conditions. Mitochondria fusion relies on the concerted action of two distinct GTPase family proteins, mitofusins (MFN1/2) and optic atrophy 1 (OPA1). MFN1/2 are two GTPases that mediate the OMM fusion, whereas the fusion of inner mitochondrial membranes is mediated by cristae remodeling protein OPA-1 [28]. On the other hand, the mitochondrial fission process depends on the activity of dynamin-related protein 1 (DRP1) that is recruited on the organelle by direct binding to Fis1, an integral protein located at the OMM [29,30]. This binding allows the oligomerization of DRP1 that forms a ring structure around the mitochondria, constricting the mitochondrial outer membrane and causing the separation of the divided organelles [28]. In the yeast, the E3 ligase MDM30 ubiquitinates and degrades the mitofusin Fzo1p, thus inhibiting the mitochondrial fusion [31]. This process is finely counter regulated by two distinct ubiquitin binding proteins, UBP12 and UBP2, that were originally reported as DUBs that negatively regulate plant immunity [32]. Both DUBs are involved in the removal of ubiquitin moieties from different lysine residues of Fzo1p, resulting in mitochondrial fusion. In particular, UBP12 removes ubiquitin moieties that stabilize Fzo1p, while UBP2 removes ubiquitin moieties that direct Fzo1p to the proteasome. Accordingly, cells lacking UBP2 show fragmented and smaller mitochondria, due to the impairment of the fission machinery, while cells lacking UBP12 show increased mitochondrial fusion [33]. In mammals, mitochondrial dynamics is tightly regulated by the UPS (Figure 1). Thus, the E3 ligase MUL1 localized at OMM, coordinates the ubiquitination and proteolysis of MFN2, limiting the mitochondrial fusion [34]. Furthermore, MUL1 sumoylates and stabilizes DRP1, preventing its proteolysis and promoting mitochondrial fission [15] and apoptosis [35]. These two important functions make mitochondrial MUL1 a key player in mitochondrial dynamics with a deep impact on cell survival. The fission mechanism is also regulated by the UPS. Thus, DRP1 and hFIS1 are ubiquitinated and degraded by the OMM E3 ligase MARCH5/MITOL, preventing mitochondrial fission [36]. Moreover, the E3 ligase synviolin (SYVN1) ubiquitinates and directs DRP1 to proteasome in the human ovarian granulosa-like tumor cell line (KGN). SYVN1 levels are markedly reduced in primary granulosa cells derived from polycystic ovary syndrome patients, resulting in increased DRP1 levels and mitochondrial fission [37].

### 2.2. Mitochondrial Quality Control

The number of mitochondria is tightly regulated through two different processes that promote the formation (biogenesis) or the elimination (mitophagy) of mitochondria. The mitochondrial biogenesis is an evolutionary conserved mechanism of self-replication through which cells enhance the number of mitochondria in response to cellular stresses, such as oxidative stress, or as consequence of extracellular stimulation by hormones, growth factors or endurance training [38]. A coordinated action of mitochondrial genome replication, transcription and translation of mitochondrial and nuclear genes supports the synthesis of components of the oxidative phosphorylation machinery, tricarboxylic acid cycle, mitochondrial metabolic pathways and membranes, promoting the formation of new organelles. Depending on specific metabolic needs and environmental conditions, the number of mitochondria varies in a cell-type dependent context [39].

Mitophagy is a conserved cellular mechanism that eliminates damaged or unneeded mitochondria through the autophagy pathway, avoiding the accumulation of dysfunctional organelles [40]. It is well known that oxidative stress regulates the mitophagy pathway through UPS. In response to mitochondrial damage or hypoxia, the mitochondrial membrane potential decreases, causing the stabilization and accumulation of the PTEN induced kinase 1 (PINK1) protein at the OMM [41]. The cytosolic E3 ubiquitin ligase Parkin, then, is rapidly recruited on mitochondria by PINK1. Once at the OMM, Parkin ubiquitinates different mitochondrial targets including the voltage dependent anion-selective channel (VDAC) and mitofusins 1/2 (MFN1/2). The ubiquitin moieties bound to these substrates furnish docking sites for autophagy receptors, allowing the initiation of the mitophagy process and the consequent removal of damaged mitochondria from the cell [42]. Although the PINK1/Parkin axis is the best characterized mechanism of mitophagy control exerted by the UPS, mitophagy can occur also in cells lacking Parkin, indicating the presence of additional players and mechanisms involved in the process. Accordingly, in the last few years, other E3 ligases have been connected with the initiation of mitophagy process in response to oxidative damage. In particular, the activating molecule in Beclin 1-regulated autophagy protein 1 (AMBRA1), the cytosolic HECT, UBA and WWE domain containing E3 ubiquitin protein ligase 1 (HUWE1) and MARCH5 have been connected with connected with Parkin-independent mitophagy. Thus, in response to oxidative stress, AMBRA1 binds and recruits HUWE1 to the OMM. HUWE1, in turn, ubiquitinates and directs MFN2 to proteasome, impairing the fusion of damaged mitochondria and favoring their elimination through mitophagy [43,44]. The mitochondrial E3 ligase MARCH5 regulates hypoxia-induced mitophagy by ubiquitinating and degrading the FUN14 domain containing protein 1 (FUNDC1). Proteolysis of FUNDC1 desensitizes mitochondria to hypoxia-induced mitophagy [45].

## 3. Control of Mitochondrial Metabolism by UPS

Mitochondria are the powerhouse of the cell and through the tricarboxylic acid cycle (TCA), also known as Krebs cycle and the oxidative phosphorylation chain produce metabolites and ATP required for cell differentiation, growth and survival. The enzymes of TCA cycle and of the respiratory chain are located within mitochondrial matrix and in the inner mitochondrial membrane (IM), respectively. In the past years, the ubiquitination of inner mitochondrial proteins seemed to be extremely unlikely, due to their spatial separation from the cytoplasm. Nevertheless, recently, the ubiquitination of inner mitochondrial proteins has been experimentally proved [46,47]. Interestingly, almost half of 720 mitochondrial ubiquitinated proteins reside in mitochondrial matrix and IM [47]. Analysis conducted in yeast identified, among all the ubiquitinated proteins, important components of the mitochondrial metabolism including aldehyde dehydrogenase (ADH3p) and isocitrate dehydrogenase (IDH2p) [47]. These findings lay the foundations for further studies in yeast aimed to define the role of compartmentalized ubiquitination of mitochondria proteins in oxidative and metabolic pathways.

### 3.1. Tricarboxylic Acid Cycle

The impact of the ubiquitin system in the regulation of mitochondrial activities has been thoroughly explored in mammalian cells where mitochondrial glutamine metabolism is essential to furnish important intermediates to the TCA cycle and metabolites for the fatty acid synthesis [48,49]. Glutamine, the most abundant amino acid in biological fluids, is converted to α-ketoglutarate (αKG) within the mitochondria by sequential enzymatic steps catalyzed by the glutaminase and glutamate dehydrogenase (GDH). αKG, then, can be further oxidized to succinate by α-ketoglutarate dehydrogenase (α-KGDH) or reduced to isocitrate by the isocitrate dehydrogenase [49]. The balance between succinate and isocitrate, that is essential to control cell proliferation, is regulated by the hypoxia-induced E3 ubiquitin ligase SIAH2 [50]. Thus, in response to hypoxia, SIAH2 accumulates, ubiquitinates and degrades the oxoglutarate dehydrogenase (OGDH) subunit of the α-KGDH. As consequence, the levels of α-KG rise and the metabolic route of glutamine is switched to the fatty acid synthesis, thus sustaining cell growth under hypoxic conditions [50]. The regulation of mitochondrial activity by SIAH2 was previously demonstrated as a key mechanism to restrain oxidative respiration in the presence of low oxygen availability. In particular, under oxygen deprivation, SIAH2 ubiquitinates the mitochondrial PKA scaffold A-kinase anchor protein 1 (AKAP1) and induces its degradation through the proteasome. Downregulation of AKAP1 by SIAH2 decreases PKA signaling to mitochondria, dampening oxidative phosphorylation and promoting mitochondrial fission [51,52,53].

### 3.2. Glucose Metabolism

The C-terminal to LisH (CTLH) complex, also called the glucose-induced degradation deficient (GID) complex, is an E3 ligase complex composed of 7 subunits that is activated by high glucose levels. In yeast, it has been linked to the ubiquitination of gluconeogenic enzymes, such as fructose 1,6 bisphosphate [54]. Cryo-electron microscopy studies show that GID assembles as dimer to recognize the substrate, activates the E2 ubiquitin conjugating enzyme 8 (UBC8) and promotes the ubiquitination of its substrates [55]. Surprisingly, the UBC8 enzyme is also involved in the mitochondrial biogenesis. Quantitative proteomic analysis revealed that depletion of UBC8 results in a severe decrease in mitochondrial proteins biosynthesis, including the translocase of outer mitochondrial membrane 22 (TOM22). TOM22 is a central component of the translocase of outer membrane complex (TOM) that mediates the transport of proteins and metabolites from cytoplasm to mitochondria [56]. The incorporation of TOM22 in the TOM complex is negatively regulated by the OMM porine protein (POR). UBC8 accelerates the assembly of TOM complex by targeting POR for the GID/CTLH-mediated proteolysis. Accordingly, the depletion of UBC8 decreases TOM22 levels and impairs the OMM import system and mitochondrial biogenesis [57], connecting the intracellular glucose levels with the mitochondrial homeostasis. Very recently, the mitochondrial E3 ligase MUL1 has been directly linked to the regulation of cell metabolism, coordinating the mitochondrial activity and the bioenergetic state of the cell. In particular, MUL1 ubiquitinates and degrades proteins involved in the glycolytic pathway, including protein kinase AKT2, Unc-51 like autophagy activating kinase 1 (ULK1) and hypoxia inducible factor 1 subunit α (HIF1α). The proteolysis of these substrates promotes mitochondrial respiration and oxidative metabolism. Accordingly, the inactivation of MUL1 increases the levels of AKT2 and HIF1α, and positively impacts on glycolysis, lipid metabolism and mitochondrial anaplerotic routes, pointing to MUL1 as a major regulator of mitochondrial respiration and metabolism [58,59].

### 3.3. Electron Transport Chain

The reciprocal regulation between UPS and mitochondrial homeostasis has been elegantly demonstrated over the past years [60,61,62,63]. A certain number of E3 ligases reside on the outer mitochondrial membrane (OMM) where they poly-ubiquitinate damaged mitochondrial proteins. These, once ubiquitinated, are recognized by Cdc48/p97 complex, translocated to the cytoplasm and addressed to proteasomal degradation [64]. Interestingly, the ubiquitinated mitochondrial proteins include key components of the electron transport chain, whose inhibition or downregulation results in increased ROS production. ROS, in turn, negatively regulate the ubiquitination and proteolysis of these mitochondrial proteins, in a reciprocal feedback loop [5,65]. This particular mechanism shows how the mitochondrial oxidative stress can be regulated by the UPS, and how the modulation of proteolysis of certain substrates can protect the cell from an abnormal accumulation of ROS.

### 3.4. Mitochondrial Uncoupling Proteins

Mitochondrial uncoupling proteins (UCP) are integral inner mitochondrial membrane proteins that dissipate the energy generated by the electron transport chain [66]. The energy produced is converted to heat, regulating the thermogenesis of specific tissues such as the brown adipose tissue. UCP expression is highly regulated at transcriptional level: the decrease in temperature induces a fast induction of UCP mRNA transcription, through the activation of the cAMP signaling pathway [67]. Over the last years, post-transcriptional mechanisms that regulate UCP activity and stability emerged [68,69]. In particular, the UPS seems to be one of the main regulators of uncoupling proteins stability. As an example, the cytosolic 26S proteasome is necessary for the degradation of uncoupling proteins 2 and 3 (UCP2,3) [70]. In addition, the uncoupling protein 1(UCP1) turnover is regulated through the UPS, as demonstrated by the detection of poly-ubiquitinated UCP1 forms in cells treated with proteasome inhibitors [69]. Poly-ubiquitination induces UCP1 proteolysis, thus facilitating the process of acclimatation. In fact, cold temperatures induce the thermogenic activity of brown fat tissue, increasing the synthesis of UCPs and heat production. In course of cold acclimatation, UCPs undergo rapid turnover to preserve the mitochondrial response to changed environmental conditions [70].Thus, the UPS by regulating the stability of UCPs plays a fundamental role in the process of body cold tolerance.

## 4. UPS, Mitochondria and Cancer

Mitochondria modify their morphology and biochemical functions in response to metabolic needs, cellular damage or changes in nutrients and oxygen availability. Mitochondria can be considered as sentinel organelles that drive the cellular homeostasis by integrating different signaling input and supplying energy and metabolites under healthy conditions. Due to their role in cellular metabolism, mitochondria coordinate different molecular pathways involved in cell proliferation and survival [71]. In the mitochondrial compartments, a variety of biochemical reactions, such as the glutamine metabolism and the fatty acids oxidation, take place, providing intermediates for TCA cycle and ATP production to support cell growth. Furthermore, mitochondria are biosynthetic hubs that furnish amino acids, nucleotides and hemes for rapidly dividing cells. These processes take advantage of a highly regulated mitochondrial redox balance that finely maintains the cellular redox homeostasis through a complex system of shuttling and transport proteins that operate to avoid the accumulation of dangerous oxidative species within the organelles [71]. Stress stimuli converge on mitochondria to induce a compensatory metabolic adaptation and, in the presence of irreversible oxidative damage, activate mitochondrial apoptotic pathways. An increasing number of mitochondrial proteins has been connected to the regulation of cytosolic processes that are pivotal for the maintenance of cellular homeostasis, like the Unfolded protein response (UPR). The correct folding and modification of newly synthetized proteins rely on the activity of chaperone proteins involved in the protein quality control [72]. Proteins that do not reach the required quality standards are targeted to the endoplasmic reticulum (ER)-associated degradative system, and the excessive accumulation of misfolded proteins within the reticulum causes ER stress. Under this condition, to keep the appropriate cellular homeostasis, ER chaperones trigger an unfolded protein response (UPR). The UPR activates stress kinases that in turn induce adaptive cellular reprogramming, mitigating the stress of the secretory pathway. Accordingly, defective or abnormal activation of UPR induces apoptosis [73]. In cancer cells, accumulation of genetic mutations often induces unfolding of many cellular proteins that eventually trigger the UPR and apoptosis. To overcome this fatal fate, tumor cells activate a mechanism of resistance to ER stress and UPR that is mostly based on MARCH5-mediated ubiquitination and proteolysis of MFN2. Downregulation of MFN2 by MARCH5 in melanoma cells induces mitochondrial fission, that in turn facilitates the autophagic removal of fragmented mitochondria. The induction of mitochondrial fission and mitophagy protects cancer cells from ER stress and programmed cell death [74]. The mitochondrial E3 ligase MARCH5/MITOL has been recently shown to have an anti-apoptotic role in the ER stress-response [74]. The inositol requiring enzyme 1 α (IRE1α) is a sensor of the UPR and acts at mitochondria-associated membranes (MAMs). In the presence of misfolded proteins, IRE1α undergoes oligomerization, thus activating the apoptotic machinery. Ubiquitination of IRE1α by MARCH5/MITOL at K63 impedes its oligomerization and inhibits the apoptotic cell death [75].

The integrated stress response (ISR) is an alternative p53-independent pathway induced by nutrients (glucose or amino acid) depletion, hypoxia, genotoxic damage and the ER stress/UPR, that increases NOXA transcription. Activation of ISR by a combined treatment of cancer cells with Navitoclax, that acts through B cell lymphoma extra-large (BCLXL) blockade, and protein kinase inhibitors induces MARCH5-mediated ubiquitination and proteolysis of the BCL2 family apoptosis regulator (MCL1). Downregulation of MCL1 markedly enhances apoptotic cancer cell death [76].

Overexpression of MARCH5 has been also linked to TGFβ-induced autophagy and promotion of growth and motility of several cancer cell types [77,78]. In particular, MARCH5 is overexpressed in ovarian cancer and its levels are correlated with the correct functioning of the autophagic machinery, since downregulation of MARCH5 severely affects the autophagic flux. The transforming growth factor-β (TGF-β) is a well-known inductor of autophagy, in fact it increases the levels of the suppressor of mothers against decapentaplegic 2 (SMAD2) and of the autophagy- related protein 5 (ATG5), positively regulating the autophagic pathway. SMAD2, in turn, regulates MARCH5 expression in a feed-back negative loop that involves the microRNA miR30A. Specifically, SMAD2/3 interacts with the miR30A promoter and induce its transcription. miR30A, in turn, downregulates both MARCH5 and ATG5 [77]. A similar mechanism operates in breast cancer cells, where MARCH5 plays an oncogenic role in promoting tumor growth and metastasis, markedly affecting cancer patients survival [78].

The phosphatidylinositol 3 kinase-AKT pathway is one of the most frequently deregulated signaling pathways in human cancer. This signaling cascade can be activated by a broad variety of extracellular stimuli, such as growth factors, hormones and cytokines and regulates multiple downstream targets resulting in increased growth, survival and proliferation [79,80]. The OMM E3 ligase MUL1 ubiquitinates and degrades the protein kinase AKT. This kinase is often upregulated or mutated in cancer, favoring cancer cell growth and invasiveness [81,82]. Given its role in the ubiquitination and proteolysis of AKT, MUL1 is often downregulated in different types of cancer [83,84]. The importance of shutting down aberrant AKT in tumor cells has focused the attention on MUL1 activity and stability. MUL1 is downregulated in head and neck cancer (HNC), and also in thyroid cancer. Treatment with cisplatin (CDDP) increases MUL1 transcription in these cancer tissues. In fact, reactive oxygen species (ROS), produced in response to cisplatin (CDDP) treatment, cause the activation of the transcriptional factor fork head box O3 (FOXO3) which, in turn, activates the transcription of MUL1 gene. The CDDP treatment, thus, results in MUL1-mediated ubiquitination and proteolysis of AKT, constituting a good therapeutic loop for HNC and thyroid cancer patients [85]. A similar mechanism has been demonstrated in ovarian cancer cells where MUL1 expression is induced by the anti-cancer agent Metformin [86]. Moreover, in clear cell renal cell carcinoma (ccRCC), MUL1 levels positively correlate with the survival rate of patients [87].

Cancer cells are strictly dependent on energy and metabolites production for growth and survival [88,89,90]. In particular, cancer cells metabolize large amounts of glucose compared to non-tumoral surrounding cells. Most of the glucose used by cancer cells is converted to lactate even in the presence of physiological levels of oxygen and with no significant functional alterations or enzymatic defects in the oxidative phosphorylation machinery. This metabolic shift, also known as ‘Warburg effect’, supplies sufficient energy and metabolites to support cancer cell growth [91]. In cancer cells, a mitochondrial control mechanism underlies the metabolic switch from oxidative respiration to Warburg effect. Specifically, the oncosuppressor p53 promotes the mitochondrial respiration and the pentose phosphate pathway (PPP), through the transcriptional activation of TP53 induced glycolysis regulatory phosphatase (TIGAR). TIGAR hydrolyzes the fructose 2–6 bisphosphate (F2-6P2), a positive allosteric regulator of the glycolytic rate-limiting enzyme phosphofructokinase. By downregulating F2-6P2 levels, TIGAR inhibits glycolysis. The induction of PPP by p53 increases the levels of NADPH and reduces glutathione (GSH), providing a defense system against oxidants and decreasing ROS-mediated DNA damage [92,93,94]. To prevent the Warburg effect, p53 also activates the transcription of the E3 ligase Parkin that supports mitochondrial activity and respiration [95]. In different tumors, decreased expression of Parkin is linked to a high glucose uptake, increased glycolytic rate and impaired oxidative metabolism [96,97,98,99]. At biochemical level, Parkin deficiency reduces the levels of multiple mitochondrial proteins, including pyruvate dehydrogenase E1α1 (PDHA1), a critical enzyme that converts pyruvate into acetyl-coenzyme A. Parkin, as a direct target of p53, contributes to enhance the oxidative metabolism and reduce the Warburg effect, playing an important tumor suppressive role [95]. Tumor growth and expansion often expose cancer cells to hypoxia. Under these conditions, cancer cells rewire the mitochondrial metabolism by transcriptionally downregulating nuclear encoded mitochondrial genes (NEMGs). Mechanistically, hypoxia induces the accumulation of the E3 ligase SIAH2, which in turn ubiquitinates and degrades the nuclear respiratory factor 1 (NRF1) that acts as a positive transcriptional regulator of NEMGs [100]. This evidence further supports the concept that the dynamic interplay between deregulated UPS and mitochondrial activities may underlie cancer development and progression.

## 5. Concluding Remarks

Mitochondria are the principal providers of energy and metabolites for all cellular activities. These organelles represent the place where signaling networks and metabolic pathways converge and integrate to support differentiation, cell growth and survival. Environmental stress conditions, poisoning or genetic mutations that affect mitochondrial activities play a pathogenic role in a wide variety of human diseases. In the last years, an important influence of the ubiquitin proteasome system in the control of mitochondrial activities emerged. The fine regulation of localization, activity and turnover of mitochondrial proteins by the UPS is a key element in the maintenance of the mitochondrial homeostasis, promoting the elimination of damaged proteins and, in some instances, priming dysfunctional mitochondria to autophagic elimination (Figure 1). Alterations of this regulatory circuitry may affect mitochondrial capability to support all cellular needs and to cop the oxidative stress, leading to the programmed cell death. This is especially relevant for non-dividing cells, as neurons, where the stress-induced loss of a significant number of cells in the brain or the peripheral nervous system may give rise to neurodegenerative disorders. It is now clear that derangement of mitochondrial signaling and activities is a smart strategy used by cancer cells to rewire metabolism and divert most of the energetic needs to the production of intermediate metabolites required for cell proliferation and tumor growth. Transcriptional and post-transcriptional regulation of components of the UPS significantly contributes to the metabolic rewiring in cancer cells. Thus, changes in the expression and/or activity of UPS enzymes, adaptors and regulatory proteins, by altering the stability and activity of mitochondrial proteins, support cancer cell metabolism and tumor growth. Understanding the molecular mechanisms underlying the intricate connections between cancer cell signaling, the ubiquitin proteasome pathway and mitochondrial activities will undoubtedly contribute to designing drugs that selectively interfere with the metabolic rewiring of cancer cells, positively impacting on personalized precision medicine for cancer patients.

## Figures and Tables

**Figure 1 cells-12-00234-f001:**
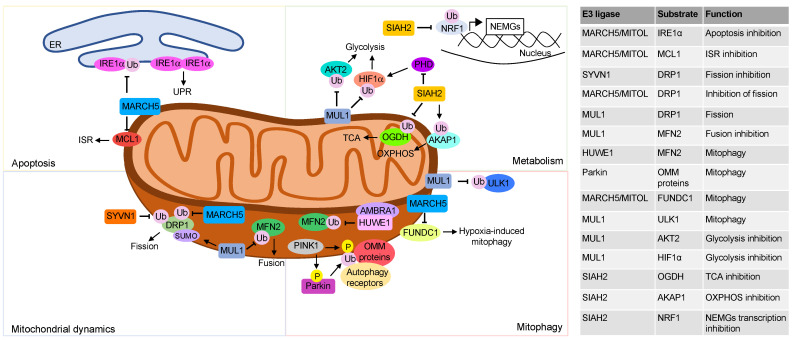
Mitochondrial regulation by UPS. The top left panel describes the ubiquitination events involving mitochondria E3 ligases that regulate apoptosis. The mitochondria E3 ligase MARCH5 ubiquitinates IRE1α located at the mitochondria associated membranes (MAMs) and inhibits apoptosis. Activation of ISR by anti-tumor drugs in solid tumors induces MARCH5-mediated ubiquitination and proteolysis of the antiapoptotic protein MCL1. The bottom left panel shows how the UPS regulates mitochondria dynamics, principally through post-translation modifications of DRP1 and MFN2. DRP1 sumoylation by MUL1 stabilizes the protein and promotes mitochondrial fission. Ubiquitin-dependent proteolysis of DRP1 by MARCH5 and SYVN1 inhibits mitochondrial fission. Moreover, the E3 ligase MUL1 by ubiquitinating and degrading MFN2 inhibits mitochondrial fusion. The top right panel describes ubiquitination events that regulate mitochondrial metabolism. The mitochondrial E3 ligase MUL1 ubiquitinates and degrades the cytoplasmatic proteins AKT2 and HIF1α rewiring cellular metabolism. The E3 ligase SIAH2 ubiquitinates and degrades the OGDH subunit of α ketoglutarate dehydrogenase, switching the glutamine metabolism to fatty acid synthesis. Moreover SIAH2, under hypoxia conditions, ubiquitinates and degrades AKAP1, reducing oxidative phosphorylation. Finally, SIAH2 ubiquitinates the transcription factor NRF1 and inhibits the transcription of the nuclear encoded mitochondrial genes (NEMGs). The bottom right panel shows how UPS at mitochondrial compartment regulates mitophagy. In response to mitochondrial damage, PINK1 accumulates at the OMM, phosphorylates ubiquitin, and recruits and activates Parkin. Parkin ubiquitinates different proteins at the outer mitochondrial membrane that are, then, recognized by autophagy receptors that promote mitophagy. Following oxidative stress, AMBRA1 recruits the E3 ligase HUWE1 at OMM. Here, HUWE1 ubiquitinates and degrades MFN2, favoring the elimination of damaged mitochondria. The mitochondrial E3 ligase MUL1 ubiquitinates ULK1 and promotes mitophagy. MARCH5 binds to- and ubiquitinates FUNDC1, promoting its proteasomal degradation. Downregulation of FUNDC1 regulates hypoxia-induced mitophagy.

## Data Availability

Not applicable.

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
