# Peer review of "Control of Mitochondrial Activity by the Ubiquitin Code in Health and Cancer"

_cells, 2023, doi:10.3390/cells12020234_

Round 1

Reviewer 1 Report

This review summarizes how mitochondrial metabolism and activities are regulated by UPS and how deregulation of this interplay then contributes to some disorders and cancer.

This review uses a myriad of abbreviations and not always consequently. Some are not defined like PTEN. My suggestion is to have all used abbreviations defined and summarized at the beginning of the manuscript, which would allow the reader an easy access. In addition, some of the abbreviations are only used once throughout the entire manuscript (for example PTEN, MMP). There is no need to introduce an abbreviation to a term, if it is used only once or twice.

At times this review reads like the writers were lost in details. The topic in itself is already complex and the ubiquitin proteasome complex not only regulates on multiple levels, but is in itself regulated in a complex way too. For example, the sequential reactions of the TCA cycle, beginning on page 4, including mentioning the catalyzing enzymes is not really necessary. The interested reader either knows the reactions of the TCA cycle or will research them.  This review would be more fun to read if it focusses directly on those reactions that are regulated by UPS.

Figure 1 is not linked to the main text

Line 81: what are dangerous mitochondria?  

Typo in line 14 vice versa

Line 192:… “IRE1α undergoes to oligomerization”…; typo? I think the word ”to” is too much

Author Response

This review summarizes how mitochondrial metabolism and activities are regulated by UPS and how deregulation of this interplay then contributes to some disorders and cancer.

This review uses a myriad of abbreviations and not always consequently. Some are not defined like PTEN. My suggestion is to have all used abbreviations defined and summarized at the beginning of the manuscript, which would allow the reader an easy access. In addition, some of the abbreviations are only used once throughout the entire manuscript (for example PTEN, MMP). There is no need to introduce an abbreviation to a term, if it is used only once or twice.

R. Many thanks to the reviewer for his/her helpful comments. We apologize for the confusion. According to the Reviewer’s suggestion, we included a list of abbreviation at the beginning of the manuscript. The abbreviations used once were eliminated and the corresponding terms were reported in full.

At times this review reads like the writers were lost in details. The topic in itself is already complex and the ubiquitin proteasome complex not only regulates on multiple levels, but is in itself regulated in a complex way too. For example, the sequential reactions of the TCA cycle, beginning on page 4, including mentioning the catalyzing enzymes is not really necessary. The interested reader either knows the reactions of the TCA cycle or will research them.  This review would be more fun to read if it focusses directly on those reactions that are regulated by UPS.

R. We thank the Reviewer for the useful advice. As suggested, we have eliminated the biochemical details of the TCA cycle and simplified the text, accordingly.

Figure 1 is not linked to the main text

R. We now remand to the Fig1, both in the first paragraph and in the Discussion section.

Line 81: what are dangerous mitochondria? 

R. The statement “dangerous mitochondria” has been eliminated. 

Typo in line 14 vice versa

R. The typo was corrected

Line 192:… “IRE1α undergoes to oligomerization”…; typo? I think the word ”to” is too much

R. We eliminated the word “to”.

Reviewer 2 Report

In this manuscript, the authors provide a concise review on the roles of ubiquitination pathways in the regulation of mitochondria function in health and cancer. This review is certainly useful for readers who would like to learn the role of ubiquitination in mitochondrial biology because it provides a broad coverage of studies on this topic. Most of the references are correctly cited. The main findings of cited studies are clearly described. However, the clarity of the presentation should be improved by organizing materials according to the pathways and biological processes discussed rather than individual proteins, such as ligases. The significance of cited studies should also be highlighted whenever possible rather than providing only a plain description of major findings in the cited studies. The many typos/grammatical errors should also be corrected before the manuscript can be accepted.

For the review to be useful for readers who work in this specific field of research, the authors should include some discussions on the future research directions in the field and any challenges, technical or conceptual, that need to be overcome by researchers in this field.

Below are my specific suggestions organized according to sections.

In “1. Introduction”

Line 32, the authors need to clarify what is “mitochondrial dynamics”. The authors mostly focused on the ligases located in mitochondria. It would be helpful to mention representative DUBs and other components of UPS such as E2, so that the readers can appreciate that the extent of involvement of various components of UPS in mitochondria function.

In “2. Ubiquitin proteasome system (UPS) in mitochondrial homeostasis”

Much of the discussion in this section focuses on mitophagy, mitochondria fusion and fission. The authors should include a short paragraph introducing these processes and the biological context within which these processes take place at the very beginning of this section to benefit readers who are not familiar with these processes.

The feed-back control between mitochondrial stress and UPS was briefly described (Lines 68-71). The authors should expand the discussion to allow the authors to at least gain some knowledge on how ROS downregulates UPS activity and the biological importance of this regulation mechanism.

In “3. “Control of mitochondria metabolism by UPS”

The authors should separate the text into multiple paragraphs according to the topics. For example, separating amino acid metabolism, glucose metabolism and protein synthesis into different paragraphs would improve the clarity of presentation.

In “4. Mitochondrial proteostasis in cancer”

The authors should include a short description on commonly deregulated pathways and biological processes in cancer, which include but are not limited to UPR, ER-stress response, kinase signaling, glucose metabolism, in the first paragraph of this section before discussing in details the functional roles of UPS components in these pathways and how these pathways are deregulated. Also, consider adding one or two sentences at the beginning of each paragraph describing the pathway(s) to be discussed in this paragraph. Consider discussing only one pathway or process in each paragraph unless there is crosstalk between different pathways.  

A few problems with references

1.    Reference 7 is incorrect

2.    Lines 37-39, a reference needs to be added for the sentence “Ubiquitination of proteins is an essential mechanism to control …”

Numerous typos/grammatical errors need to be corrected. Below is a partial list of these errors.

Line 10, change “undergoing to a programmed death” to “undergoing programmed death”

Lines 13-14, change “a pathological condition” to “pathological conditions”; change “viceversa” to “vice versa” or “conversely”

Line 30, change “principle source” to “principal source”

Line 36, change “evidences have shown” to “evidence has shown”

Line 129, need to clarify what was happening in “In the past years”

Line 161, change “fructose 1-6 bisphosphate” to “fructose 1,6 bisphosphate”

Line 192, change “undergoes to” to “undergoes”

Line 240, change “Trp53” to “TP53”

Line 273, change “cop” to “cope”; change “expecially” to “especially”

Line 285, change “undoubtly” to “undoubtedly”; change “to design” to “to designing”

Line 286, change “a personalized medicine” to “personalized medicine”

Author Response

In this manuscript, the authors provide a concise review on the roles of ubiquitination pathways in the regulation of mitochondria function in health and cancer. This review is certainly useful for readers who would like to learn the role of ubiquitination in mitochondrial biology because it provides a broad coverage of studies on this topic. Most of the references are correctly cited. The main findings of cited studies are clearly described. However, the clarity of the presentation should be improved by organizing materials according to the pathways and biological processes discussed rather than individual proteins, such as ligases. The significance of cited studies should also be highlighted whenever possible rather than providing only a plain description of major findings in the cited studies. The many typos/grammatical errors should also be corrected before the manuscript can be accepted.

For the review to be useful for readers who work in this specific field of research, the authors should include some discussions on the future research directions in the field and any challenges, technical or conceptual, that need to be overcome by researchers in this field.

Below are my specific suggestions organized according to sections.

In “1. Introduction”

Line 32, the authors need to clarify what is “mitochondrial dynamics”. The authors mostly focused on the ligases located in mitochondria. It would be helpful to mention representative DUBs and other components of UPS such as E2, so that the readers can appreciate that the extent of involvement of various components of UPS in mitochondria function.

 R. We thank the Reviewer for his/her insightful suggestions. We have now explained the sentence “mitochondrial dynamics” as “Mitochondrial fusion and fission”. DUBS and E2 enzymes were mentioned in the introduction (lines 80 to 86).

In “2. Ubiquitin proteasome system (UPS) in mitochondrial homeostasis”

Much of the discussion in this section focuses on mitophagy, mitochondria fusion and fission. The authors should include a short paragraph introducing these processes and the biological context within which these processes take place at the very beginning of this section to benefit readers who are not familiar with these processes.

R. A more exhaustive explanation about mitochondrial dynamics has now been included (lines 99 to 111; 133-146). Moreover, the paragraph 2 has been divided in subsections (2.1. Mitochondria dynamics; 2.2. Mitochondrial quality control).

The feed-back control between mitochondrial stress and UPS was briefly described (Lines 68-71). The authors should expand the discussion to allow the authors to at least gain some knowledge on how ROS downregulates UPS activity and the biological importance of this regulation mechanism.

R. The section describing the regulation of UPS by ROS has now been expanded and it is described in the paragraph 3, (subsection 3.3., lines 231-240).

In “3. “Control of mitochondria metabolism by UPS”

The authors should separate the text into multiple paragraphs according to the topics. For example, separating amino acid metabolism, glucose metabolism and protein synthesis into different paragraphs would improve the clarity of presentation.

R. The text has been separated in 4 different subections: 3.1. Tricarboxylic acid cycle; 3.2. Glucose metabolism; 3.3. Electron transport chain; 3.4. Mitochondrial ucoupling proteins.

In “4. Mitochondrial proteostasis in cancer”

The authors should include a short description on commonly deregulated pathways and biological processes in cancer, which include but are not limited to UPR, ER-stress response, kinase signaling, glucose metabolism, in the first paragraph of this section before discussing in details the functional roles of UPS components in these pathways and how these pathways are deregulated.

R. An explanatory paragraph has been now added to better describe the importance of mitochondria function for cellular homeostasis (lines 264-286).

Also, consider adding one or two sentences at the beginning of each paragraph describing the pathway(s) to be discussed in this paragraph. Consider discussing only one pathway or process in each paragraph unless there is crosstalk between different pathways.

R. Sentences describing the pathways discussed in the paragraph have been added (lines 319-323; 338-344). 

A few problems with references

  1. Reference 7 is incorrect

R. The reference has been corrected.

  1. Lines 37-39, a reference needs to be added for the sentence “Ubiquitination of proteins is an essential mechanism to control …”

Reference 7,8,9 have been added to describe the sentence.

Numerous typos/grammatical errors need to be corrected. Below is a partial list of these errors.

Line 10, change “undergoing to a programmed death” to “undergoing programmed death”

Lines 13-14, change “a pathological condition” to “pathological conditions”; change “viceversa” to “vice versa” or “conversely”

Line 30, change “principle source” to “principal source”

Line 36, change “evidences have shown” to “evidence has shown”

Line 129, need to clarify what was happening in “In the past years”

Line 161, change “fructose 1-6 bisphosphate” to “fructose 1,6 bisphosphate”

Line 192, change “undergoes to” to “undergoes”

Line 240, change “Trp53” to “TP53”

Line 273, change “cop” to “cope”; change “expecially” to “especially”

Line 285, change “undoubtly” to “undoubtedly”; change “to design” to “to designing”

Line 286, change “a personalized medicine” to “personalized medicine”

R. All the typos have been corrected, accordingly.

Reviewer 3 Report

Control of mitochondrial activity by the ubiquitin code in health and cancer” By Laura Rinaldi et al.

This review paper gives an overview of the important reciprocal regulation between the complex ubiquitin proteasome system (UPS) and mitochondrial homeostasis in health and cancer, with particular attention on how deregulation or mutations in this control mechanism result in a progression of proliferative disorder.

Overall, this review paper is interesting and informative and therefore deserves to be published.

Minor Suggestions:

- For a better understanding of the mechanisms involved in the metabolic switch in cancer cells I recommend the authors to add a summary figure, corresponding to section 4, “Mitochondrial proteostasis in cancer”, lines 234 to line 253.

-   It would be helpful if the authors could somehow add an overview of all abbreviations

Author Response

This review paper gives an overview of the important reciprocal regulation between the complex ubiquitin proteasome system (UPS) and mitochondrial homeostasis in health and cancer, with particular attention on how deregulation or mutations in this control mechanism result in a progression of proliferative disorder. Overall, this review paper is interesting and informative and therefore deserves to be published.

R. Many thanks to the Reviewer to support the publication of our review article.

Minor Suggestions:

- For a better understanding of the mechanisms involved in the metabolic switch in cancer cells I recommend the authors to add a summary figure, corresponding to section 4, “Mitochondrial proteostasis in cancer”, lines 234 to line 253.

R. The mechanisms described in lines 234-253 have been included in Figure 1.

-   It would be helpful if the authors could somehow add an overview of all abbreviations

R. A list of abbreviation has been included at the beginning of the manuscript.

Round 2

Reviewer 1 Report

The changes the authors made to this manuscript have greatly improved its quality.

Reviewer 2 Report

The authors have addressed all of my concerns in the revised manuscript.